# Online EXP3 Learning in Adversarial Bandits with Delayed Feedback

**Ilai Bistritz[1], Zhengyuan Zhou[23], Xi Chen[2], Nicholas Bambos[1], Jose Blanchet[1]**
[1]Stanford University
[2]New York University, Stern School of Business
[3]IBM Research
{bistritz,bambos,jose.blanchet}@stanford.edu, {zzhou,xchen3}@stern.nyu.edu

## Abstract

Consider a player that in each of $T$ rounds chooses one of $K$ arms. An adversary chooses the cost of each arm in a bounded interval, and a sequence of feedback delays $\{d_t\}$ that are unknown to the player. After picking arm $a_t$ at round $t$, the player receives the cost of playing this arm $d_t$ rounds later. In cases where $t + d_t > T$, this feedback is simply missing. We prove that the EXP3 algorithm (that uses the delayed feedback upon its arrival) achieves a regret of $O\left(\sqrt{\ln K \left(KT + \sum_{t=1}^{T} d_t\right)}\right)$. For the case where $\sum_{t=1}^{T} d_t$ and $T$ are unknown, we propose a novel doubling trick for online learning with delays and prove that this adaptive EXP3 achieves a regret of $O\left(\sqrt{\ln K \left(K^2 T + \sum_{t=1}^{T} d_t\right)}\right)$. We then consider a two player zero-sum game where players experience asynchronous delays. We show that even when the delays are large enough such that players no longer enjoy the "no-regret property", (e.g., where $d_t = O\left(t \log t\right)$) the ergodic average of the strategy profile still converges to the set of Nash equilibria of the game. The result is made possible by choosing an adaptive step size $\eta_t$ that is not summable but is square summable, and proving a "weighted regret bound" for this general case.

## 1 Introduction

Consider an agent that makes $T$ sequential decisions from a set of $K$ options (i.e., arms), where each decision incurs some cost. The cost sequences are chosen by an adversary that knows the agent's strategy. The agent's goal is to minimize this cost over time. In the full information case the agent gets to know the cost of all arms after choosing a single arm. A more challenging case is the bandit feedback one, where the agent only observes the cost of the chosen arm. In this paper, we consider the bandit feedback case. The question of **what** the agent learns about the costs (i.e., full information or bandit) naturally influences the best performance the agent can guarantee. Another fundamental question is **when** the agent gets to know the cost.

An online learning scenario with no delays means that the agent always knows how beneficial all the past actions were when making the current decision. This is rarely the case in practice, where many decisions have to be made before all the feedback from past choices is received. Determining the feedback in practice is not always straightforward and might involve some computations and estimations. Furthermore, the time it takes to receive the feedback varies between different decisions and times. All of these effects are accentuated when an adversary has control over the feedback mechanism. Following this reasoning, online learning with delayed feedback has attracted consid-

erable attention [1–12]. The concept of adversarial delays (i.e., arbitrary delay sequences) was first introduced in [13], for the full information case and under the assumption that all feedback is received before round $T$ (which we do not make here). The first goal of this paper is to address the more challenging bandit cost scenario.

When there is no delayed feedback, EXP3 [14–16] is the state-of-the-art algorithm for adversarial online learning with bandit feedback. In EXP3, the agent keeps a weight for each arm, and picks an arm at random with a probability that is proportional to the exponents of the weights. When a cost $l^{(i)}$ is incurred for choosing arm $i$, which was picked with probability $p^{(i)}$, $\frac{l^{(i)}}{p^{(i)}}$ is added to the weight of this arm. The idea is that on average over the randomness of the decisions, the weights are adjusted with the vector of costs $\left(l^{(1)}, ..., l^{(K)}\right)$. With no delays, the expected regret of EXP3 is $O\left(\sqrt{TK \ln K}\right)$. Having a sublinear regret, the average regret per round goes to zero as $T \to \infty$, which is known as the "no-regret property" [17].

Our first main contribution in this paper is to show that with an arbitrary sequence of delays $d_t$, EXP3 achieves an expected regret of $O\left(\sqrt{\ln K \left(KT + \sum_{t \notin \mathcal{M}} d_t\right)} + |\mathcal{M}|\right)$, where $\mathcal{M}$ is the set of rounds whose feedback is not received before round $T$. This expression makes clear which delay sequences will maintain the no-regret property and which will lead to linear regret in $T$.

An omnipotent adversary represents the embodiment of the agent's worst fears when learning to optimize its decisions in an unknown environment. An algorithm with performance guarantees in this worst case scenario is an appealing choice from a designer's point of view. As such, it is more likely that the opponents that the agent will face are online learning agents like itself, which have limited knowledge and power. These agents have interests of their own, but in the worst case these interests are in a direct conflict with those of our agent. Therefore, zero-sum games are the natural framework to analyze the outcome of an interaction against another agent instead of against an all powerful adversary. Interestingly enough, it turns out that with delayed feedback, the outcome of playing against another agent can be essentially different from playing against an adversary.

It is well known that when two agents use a no-regret learning algorithm against each other in a zero-sum game, the dynamics will result in a Nash equilibrium (NE) [18]. To be precise, the ergodic average strategy converges to the set of NE strategies and the ergodic average cost to the value of the zero-sum game. The last iterate does not converge in general to a NE, and even moves away from it [19]. However, the emergence of a NE in a game where such an agent finds itself against another agent using a no-regret algorithm provides yet another strong evidence for the importance of the concept of NE. From a more practical point of view, convergence of the ergodic average to a NE makes no-regret algorithms an appealing way to compute a NE when the game matrix is unknown and only simulating the game is possible. In such a simulation of an unknown game, bandit feedback is a more realistic assumption than full information.

With no delays, the only purpose of the step size of the EXP3 algorithm is to minimize the regret. If the horizon of the game $T$ is unknown, one can use the doubling trick and choose the step sizes accordingly. With delayed feedback, a varying step-size plays a much more central role. With delayed feedback, it is not surprising that convergence of the ergodic average to the set of NE is maintained if the algorithm still has a sublinear regret (asymptotically zero average regret). When the delays become larger, for example super-linear delays that grow like $O\left(t \log t\right)$, this is no longer true and the regret of EXP3 (or any other algorithm) becomes linear in the horizon $T$.

Our second main contribution in this paper is to show that even with delays that cause a linear regret, the ergodic average may still converge to the set of NE by using a time-varying step size $\eta_t$. This means that computing a NE using EXP3 is still possible even in scenarios where EXP3 does not enjoy a sublinear regret (i.e., the no-regret). Since delays are a prominent feature of almost every computational environment, this is an encouraging finding.

## 2  EXP3 in Adversarial Bandits under Feedback Delays

Consider a player that at each round $t$ has to pick one out of $K$ arms. Denote the arm the player chooses at round $t$ by $a_t$. The cost at round $t$ from arm $i$ is $l_t^{(i)} \in [0, 1]$, and let $\boldsymbol{l}_t = \left(l_t^{(1)}, ..., l_t^{(K)}\right)$ be the cost vector. These costs are arbitrarily chosen by an adversary that knows the player's strategy

**Algorithm 1** EXP3 with delays

---

**Initialization:** Let $\{\eta_t\}$ be a positive non-increasing sequence, and set $\tilde{L}_1^{(i)} = 0$ and $p_1^{(i)} = \frac{1}{K}$ for $i = 1, ..., K$.

**For** $t = 1, ..., T$ **do**

  1. Choose an arm $a_t$ at random according to the distribution $\boldsymbol{p}_t$.

  2. Obtain a set of delayed costs $l_s^{(a_s)}$ for all $s \in \mathcal{S}_t$, where $a_s$ is the arm played at round $s$.

  3. Update the weights of arm $a_s$ for all $s \in \mathcal{S}_t$, using

$$\tilde{L}_t^{(a_s)} = \tilde{L}_{t-1}^{(a_s)} + \eta_s \frac{l_s^{(a_s)}}{p_s^{(a_s)}}. \tag{3}$$

  4. Update the mixed strategy

$$p_{t+1}^{(i)} = \frac{e^{-\tilde{L}_t^{(i)}}}{\sum_{j=1}^{n} e^{-\tilde{L}_t^{(j)}}}. \tag{4}$$

**End**

---

in advance. Hence, we can assume that the adversary chooses $\left\{ l_t^{(i)} \right\}_t$ for each $i$ in advance, knowing exactly how the player is going to react. The player gets to know the cost of playing $a_t$ at round $t$ at the end of the $t + d_t - 1$ round (i.e., after a delay of $d_t \geq 1$ rounds), so the feedback is available at the beginning of round $t + d_t$. The set of costs (feedback samples) received at round $t$ is denoted $\mathcal{S}_t$, so $s \in \mathcal{S}_t$ means that the cost of $a_s$ from round $s$ is received at round $t$. Since the game lasts for $T$ rounds, all costs for which $t + d_t > T$ are never received. Of course, the value of $d_t$ does not matter as long as $t + d_t > T$, and these are just samples that the adversary chose to prevent the player from receiving. We name these costs the missing samples, and denote their set by $\mathcal{M}$.

The player wants to have a learning algorithm that uses the past observations to make good decisions over time. Denote the vector of probabilities of the player for choosing arms at round $t$ by $\boldsymbol{p}_t \in \Delta^K$, where $\Delta^K$ denotes the $K$-simplex. This is also known as the mixed strategy of the player. The performance of the player's algorithm, or strategy, is measured using the regret. The expected regret is the total expected cost over an horizon of $T$ rounds, compared to the total cost that would result from playing the best fixed mixed strategy in all rounds:

**Definition 1.** The expected regret is defined as:

$$E^{\boldsymbol{a}}\{R(T)\} = E^{\boldsymbol{a}}\left\{ \sum_{t=1}^{T} l_t^{(a_t)} - \min_i \sum_{t=1}^{T} l_t^{(i)} \right\} \tag{1}$$

where $E^{\boldsymbol{a}}$ is the expectation over the random actions $a_1, ..., a_T$ the agent chooses at each round.

At round $t$, EXP3 (detailed in Algorithm 1) chooses an arm at random according to the distribution $\boldsymbol{p}_t$ that depends on the history of the game. Define the following filtration

$$\mathcal{F}_t = \sigma\left( \{a_s \mid s + d_s \leq t\} \right) \tag{2}$$

which is generated from all the actions for which the feedback was received up to round $t$. Note that the mixed strategy of the player $\boldsymbol{p}_t$ is a $\mathcal{F}_t$-measurable random variable, since $\boldsymbol{p}_t$ is a function of all feedback received up to round $t$.

Our main result of this section establishes the expected regret bound for EXP3 with delays. Note that Algorithm 1 is nothing but the obvious variant of EXP3 for the case of delayed feedback. Therefore, the importance of the following result is in the novel analysis of how delays, which are a part of every practical system, affect a well-known and widely used algorithm such as EXP3. While waiting for the delayed feedback, the agent is making decisions that incur a larger regret than in the usual no-delay case where all the past feedback has been received. The proof of Theorem 1 bounds this addition to the regret. The proof analyzes the novel notion of weighted-regret, given in the following Lemma. The goal of this more general result is to be both used here and for the proof of Theorem 3 in the next section.

**Lemma 1.** *Let $\{\eta_t\}$ be a non-increasing step size sequence. Let $\left\{l_t^{(i)}\right\}$ be a cost sequence such that $l_t^{(i)} \in [0, 1]$ for every $t, i$. Let $\{d_t\}$ be a delay sequence such that the cost from round $t$ is received at round $t + d_t$. Define $\mathcal{M}$ to be the set of all samples that are not received before round $T$. Then using EXP3 (Algorithm 1) guarantees*

$$E^{\boldsymbol{a}}\left\{\sum_{t=1}^{T}\eta_t l_t^{(a_t)} - \min_i \sum_{t=1}^{T}\eta_t l_t^{(i)}\right\} \leq \ln K + \frac{K}{2}\sum_{t=1}^{T}\eta_t^2 + 4\sum_{t\notin\mathcal{M}}\eta_t^2 d_t + \sum_{t\in\mathcal{M}}\eta_t. \qquad (5)$$

*Proof.* Let $\boldsymbol{e}_i$ be the pure strategy that picks arm $i$ with probability 1. Then for each $i$

$$E^{\boldsymbol{a}}\left\{\sum_{t=1}^{T}\eta_t l_t^{(a_t)} - \sum_{t=1}^{T}\eta_t l_t^{(i)}\right\} = E^{\boldsymbol{a}}\left\{\sum_{t=1}^{T}E^{\boldsymbol{a}}\left\{\eta_t l_t^{(a_t)}\mid\mathcal{F}_t\right\} - \sum_{t=1}^{T}\eta_t l_t^{(i)}\right\} =$$

$$E^{\boldsymbol{a}}\left\{\sum_{t=1}^{T}\eta_t\left\langle l_t,\boldsymbol{p}_t\right\rangle - \sum_{t=1}^{T}\eta_t l_t^{(i)}\right\} = E^{\boldsymbol{a}}\left\{\sum_{t=1}^{T}\eta_t\left\langle l_t,\boldsymbol{p}_t - \boldsymbol{e}_i\right\rangle\right\} =$$

$$E^{\boldsymbol{a}}\left\{\sum_{t=1}^{T}\sum_{s\in\mathcal{S}_t}\eta_s\left\langle l_s,\boldsymbol{p}_s - \boldsymbol{e}_i\right\rangle\right\} + E^{\boldsymbol{a}}\left\{\sum_{t\in\mathcal{M}}\eta_t\left\langle l_t,\boldsymbol{p}_t - \boldsymbol{e}_i\right\rangle\right\} \underset{(a)}{\leq}$$

$$E^{\boldsymbol{a}}\left\{\sum_{t=1}^{T}\sum_{s\in\mathcal{S}_t}\eta_s\left\langle l_s,\boldsymbol{p}_s - \boldsymbol{e}_i\right\rangle\right\} + \sum_{t\in\mathcal{M}}\eta_t \qquad (6)$$

where (a) follows from $\left\langle l_t,\boldsymbol{p}_t - \boldsymbol{e}_i\right\rangle \leq 1$, since $0 \leq l_t^{(i)} \leq 1$ for every $i$.

Define $\mathcal{S}_{t,s} = \{r \in \mathcal{S}_t; r < s\}$. This is the set of feedback samples arriving at round $t$ that the algorithm uses before $s$. Define $s_-$ as the step a moment before using the feedback from round $s$, so $\boldsymbol{p}_{s_-}$ is the mixed strategy at this moment. Define $s_+$ as the step a moment after using the feedback from round $s$. This step is taking place in round $t$ if $s \in \mathcal{S}_t$. We analyze the first term in (6) by splitting it as follows

$$E^{\boldsymbol{a}}\left\{\sum_{t=1}^{T}\sum_{s\in\mathcal{S}_t}\eta_s\left\langle l_s,\boldsymbol{p}_s - \boldsymbol{e}_i\right\rangle\right\} = E^{\boldsymbol{a}}\left\{\sum_{t=1}^{T}\sum_{s\in\mathcal{S}_t}\eta_s\left\langle l_s,\boldsymbol{p}_{s_-} - \boldsymbol{e}_i\right\rangle + \sum_{t=1}^{T}\sum_{s\in\mathcal{S}_t}\eta_s\left\langle l_s,\boldsymbol{p}_s - \boldsymbol{p}_{s_-}\right\rangle\right\}$$
$$\qquad (7)$$

where the first part is interpreted as the regret with no delays, and the second as the regret penalty the delays incur. From Lemma 3 we have

$$E^{\boldsymbol{a}}\left\{\sum_{t=1}^{T}\sum_{s\in\mathcal{S}_t}\eta_s\left\langle l_s,\boldsymbol{p}_{s_-}\right\rangle - \sum_{t=1}^{T}\eta_t l_t^{(i)}\right\} \leq \ln K + \frac{K}{2}\sum_{t=1}^{T}\eta_t^2. \qquad (8)$$

Next we analyze the delay term. Let $\tilde{l}_t = \left(0,...,\frac{l_t^{(a_t)}}{p_t^{(a_t)}},...,0\right)$. First note that for all $i$ we have

$$p_{q_-}^{(i)} = \frac{e^{-\tilde{L}_{q_-}^{(i)}}}{\sum_{j=1}^{K}e^{-\tilde{L}_{q_-}^{(j)}}} \triangleq h_i\left(\tilde{\boldsymbol{L}}_q\right) \qquad (9)$$

and $p_{q_+}^{(i)} = h_i\left(\tilde{\boldsymbol{L}}_{q_-} + \eta_q\tilde{l}_q\right)$, so from Lemma 2 using $\boldsymbol{x} = \tilde{\boldsymbol{L}}_{q_-}$ and $\Delta = \eta_q\tilde{l}_q$, so $\boldsymbol{h}(\boldsymbol{x}) = \boldsymbol{p}_{q_-}$ we obtain

$$E^{\boldsymbol{a}}\left\{\left\|\boldsymbol{p}_{q_+} - \boldsymbol{p}_{q_-}\right\|_1\mid\mathcal{F}_{q_-}\right\} \leq 2\eta_q E^{\boldsymbol{a}}\left\{\sum_{i=1}^{K}p_{q_-}^{(i)}\tilde{l}_q^{(i)}\mid\mathcal{F}_{q_-}\right\} \underset{(a)}{=}$$

$$2\eta_q\sum_{i=1}^{K}p_{q_-}^{(i)}E^{\boldsymbol{a}}\left\{\tilde{l}_q^{(i)}\mid\mathcal{F}_{q_-}\right\} \underset{(b)}{=} 2\eta_q\sum_{i=1}^{K}p_{q_-}^{(i)}l_q^{(i)} \leq 2\eta_q\sum_{i=1}^{K}p_{q_-}^{(i)} = 2\eta_q \qquad (10)$$

where (a) uses $p_{q_-}^{(i)} \in \mathcal{F}_{q_-}$ and (b) uses $p_q^{(i)} \in \mathcal{F}_{q_-}$ (since $q < q_-$) together with the fact that $\tilde{l}_q^{(i)}$ is $\frac{l_q^{(i)}}{p_q^{(i)}}$ with probability $p_q^{(i)}$ and zero otherwise. Note that $a_q$ is independent of $\mathcal{F}_{q_-}$ since by definition the feedback from $a_q$ was not received until round $q_-$. Therefore

$$E^{\boldsymbol{a}} \left\{ \sum_{t=1}^{T} \sum_{s \in \mathcal{S}_t} \eta_s \left\langle \boldsymbol{l}_s, \boldsymbol{p}_s - \boldsymbol{p}_{s_-} \right\rangle \right\} =$$

$$E^{\boldsymbol{a}} \left\{ \sum_{t=1}^{T} \sum_{s \in \mathcal{S}_t} \eta_s \left( \left\langle \boldsymbol{l}_s, \boldsymbol{p}_t - \boldsymbol{p}_{s_-} \right\rangle + \sum_{r=s}^{t-1} \left\langle \boldsymbol{l}_s, \boldsymbol{p}_r - \boldsymbol{p}_{r+1} \right\rangle \right) \right\} =$$

$$E^{\boldsymbol{a}} \left\{ \sum_{t=1}^{T} \sum_{s \in \mathcal{S}_t} \eta_s \left( \left\langle \boldsymbol{l}_s, \sum_{q \in \mathcal{S}_{t,s}} (\boldsymbol{p}_{q_-} - \boldsymbol{p}_{q_+}) \right\rangle + \sum_{r=s}^{t-1} \left\langle \boldsymbol{l}_s, \sum_{q \in \mathcal{S}_r} (\boldsymbol{p}_{q_-} - \boldsymbol{p}_{q_+}) \right\rangle \right) \right\} \underset{(a)}{\leq}$$

$$E^{\boldsymbol{a}} \left\{ \sum_{t=1}^{T} \sum_{s \in \mathcal{S}_t} \eta_s \left( \|\boldsymbol{l}_s\|_\infty \left\| \sum_{q \in \mathcal{S}_{t,s}} (\boldsymbol{p}_{q_+} - \boldsymbol{p}_{q_-}) \right\|_1 + \sum_{r=s}^{t-1} \|\boldsymbol{l}_s\|_\infty \left\| \sum_{q \in \mathcal{S}_r} (\boldsymbol{p}_{q_+} - \boldsymbol{p}_{q_-}) \right\|_1 \right) \right\} \underset{(b)}{\leq}$$

$$E^{\boldsymbol{a}} \left\{ \sum_{t=1}^{T} \sum_{s \in \mathcal{S}_t} \eta_s \left( \sum_{q \in \mathcal{S}_{t,s}} \|\boldsymbol{p}_{q_+} - \boldsymbol{p}_{q_-}\|_1 + \sum_{r=s}^{t-1} \sum_{q \in \mathcal{S}_r} \|\boldsymbol{p}_{q_+} - \boldsymbol{p}_{q_-}\|_1 \right) \right\} =$$

$$E^{\boldsymbol{a}} \left\{ \sum_{t=1}^{T} \sum_{s \in \mathcal{S}_t} \eta_s \sum_{q \in \mathcal{S}_{t,s}} E \left\{ \|\boldsymbol{p}_{q_+} - \boldsymbol{p}_{q_-}\|_1 \mid \mathcal{F}_{q_-} \right\} \right\} +$$

$$E^{\boldsymbol{a}} \left\{ \sum_{t=1}^{T} \sum_{s \in \mathcal{S}_t} \eta_s \sum_{r=s}^{t-1} \sum_{q \in \mathcal{S}_r} E \left\{ \|\boldsymbol{p}_{q_+} - \boldsymbol{p}_{q_-}\|_1 \mid \mathcal{F}_{q_-} \right\} \right\} \underset{(c)}{\leq}$$

$$2 E^{\boldsymbol{a}} \left\{ \sum_{t=1}^{T} \sum_{s \in \mathcal{S}_t} \eta_s \left( \sum_{q \in \mathcal{S}_{t,s}} \eta_q + \sum_{r=s}^{t-1} \sum_{q \in \mathcal{S}_r} \eta_q \right) \right\} \underset{(d)}{\leq} 4 \sum_{t \notin \mathcal{M}} \eta_t^2 d_t \quad (11)$$

where (a) follows from Hölder's inequality, (b) since $\left| l_t^{(i)} \right| \leq 1$ for every $i$ and using the triangle inequality, (c) from (10) and (d) follows from Lemma 4.

Combining (6), (8) and (11) yields, for all $i = 1, ..., K$

$$E^{\boldsymbol{a}} \left\{ \sum_{t=1}^{T} \eta_t l_t^{(a_t)} - \sum_{t=1}^{T} \eta_t l_t^{(i)} \right\} \leq \ln K + \frac{K}{2} \sum_{t=1}^{T} \eta_t^2 + 4 \sum_{t \notin \mathcal{M}} \eta_t^2 d_t + \sum_{t \in \mathcal{M}} \eta_t. \quad (12)$$

$\square$

**Theorem 1.** *Define $\mathcal{M}$ to be the set of all samples that are not received before round $T$. Choose the fixed step size $\eta = \sqrt{\frac{\ln K}{KT + \sum_{t \notin \mathcal{M}} d_t}}$. Let $\left\{ l_t^{(i)} \right\}$ be a cost sequence such that $l_t^{(i)} \in [0, 1]$ for every $t, i$. Let $\{d_t\}$ be a delay sequence such that the cost from round $t$ is received at round $t + d_t$. Then*

$$E^{\boldsymbol{a}} (R(T)) = E \left\{ \sum_{t=1}^{T} l_t^{(a_t)} - \min_i \sum_{t=1}^{T} l_t^{(i)} \right\} \leq O \left( \sqrt{\ln K \left( KT + \sum_{t \notin \mathcal{M}} d_t \right)} + |\mathcal{M}| \right). \quad (13)$$

*Proof of Theorem 1.* To obtain Theorem 1, substitute $\eta_t = \eta$ in (5) of Lemma 1, and divide both sides by $\eta$:

$$E^{\boldsymbol{a}} \left\{ \sum_{t=1}^{T} l_t^{(a_t)} - \min_i \sum_{t=1}^{T} l_t^{(i)} \right\} \leq O \left( \frac{\ln K}{\eta} + \eta \left( KT + \sum_{t \notin \mathcal{M}} d_t \right) + |\mathcal{M}| \right) \quad (14)$$

Then, choosing $\eta = \sqrt{\frac{\ln K}{KT + \sum_{t \notin \mathcal{M}} d_t}}$ yields (13). $\square$

It is worthwhile noting that our bound is tighter than $O\left(\sqrt{\ln K\left(KT + \sum_{t=1}^{T} d_t\right)}\right)$ that does not take $\mathcal{M}$ into account, since counting delays that go beyond round $T$ is redundant. For example, if $d_t = t^2$ then $\sqrt{\sum_{t=1}^{T} d_t} = O\left(T^{\frac{3}{2}}\right)$. Our subsequent Corollary formalizes this intuition.

**Corollary 1.** *Let* $\eta = \sqrt{\frac{\ln K}{KT + \sum_{t \notin \mathcal{M}} d_t}}$. *Let* $\left\{l_t^{(i)}\right\}$ *be a cost sequence such that* $l_t^{(i)} \in [0,1]$ *for every* $t, i$. *Let* $\{d_t\}$ *be a delay sequence such that the cost from round* $t$ *is received at round* $t + d_t$. *Then*

$$E^{\boldsymbol{a}}\left(R\left(T\right)\right) = E^{\boldsymbol{a}}\left\{\sum_{t=1}^{T} l_t^{(a_t)} - \min_{i} \sum_{t=1}^{T} l_t^{(i)}\right\} \leq O\left(\sqrt{\ln K\left(KT + \sum_{t=1}^{T} d_t\right)}\right) \qquad (15)$$

*Proof.* The $m = |\mathcal{M}|$ missing samples (received after $T$) contribute at least $\frac{m(m+1)}{2}$ to the sum of delays $\sum_{t=1}^{T} d_t$ (since the best case is when the feedback of $T$ is delayed by one and arrives after $T$, the feedback of $T-1$ now has to be delayed by at least 2 to be received after $T$ and so on $m$ times). Hence

$$\sqrt{\ln K\left(KT + \sum_{t=1}^{T} d_t\right)} \geq \sqrt{\ln K\left(KT + \sum_{t \notin \mathcal{M}} d_t + \frac{m(m+1)}{2}\right)} \underset{(a)}{\geq}$$

$$\frac{1}{2}\sqrt{\ln K\left(KT + \sum_{t \notin \mathcal{M}} d_t\right)} + \frac{1}{2}\sqrt{\ln K \frac{m(m+1)}{2}} \geq O\left(\sqrt{\ln K\left(KT + \sum_{t \notin \mathcal{M}} d_t\right)} + |\mathcal{M}|\right)$$

$$(16)$$

where (a) follows from the concavity of $f\left(x\right) = \sqrt{x}$. $\qquad\square$

The expression in (15) reveals a robustness property of the regret bound of EXP3 under delays. While the first term in the regret, $KT \ln K$, has a factor of $K$, the delay term $\sum_{t=1}^{T} d_t$ does not have a factor of $K$. Consider bounded delays of the form $d_t = K$. Then, the order of magnitude of the regret as a function of $T$ and $K$ is $O\left(\sqrt{TK \ln K}\right)$, exactly as that of EXP3 without delays [14]. For comparison, consider the full information case where at each round the cost of all arms is received. Assume that the player uses the exponential weights algorithm, which is the equivalent of EXP3 for the full information case. For the same delay sequence $d_t = K$, exponential weights achieves a regret bound of $O\left(\sqrt{TK \ln K}\right)$ [13], $\sqrt{K}$ times worse than the $O\left(\sqrt{T \ln K}\right)$ that exponential weights with no delays achieves. The intuition for this result is that EXP3 already "paid the price" for using $K$ times less feedback than in the full information case. Depending on less feedback, EXP3 is inherently more robust to feedback delays.

## 2.1 Adaptive Algorithm: Doubling Trick with Delays

The step size $\eta = \sqrt{\frac{\ln K}{KT + \sum_{t \notin \mathcal{M}} d_t}}$ used in Algorithm 1 requires knowledge of $T$ and $\sum_{t=1}^{T} d_t$. With no delays, the standard doubling trick (see [20]) can be used if $T$ is unknown. However, the same doubling trick does not work with delayed feedback. We now present a novel doubling trick for the delayed feedback case, where $T$ **and** $\sum_{t=1}^{T} d_t$ are unknown. Define $m_t$ as the number of missing feedback samples at round $t$, starting from the $t$-th feedback. The idea is to start a new epoch every time $\sum_{\tau=1}^{t} m_\tau$, that tracks $\sum_{\tau=1}^{t} d_\tau$, doubles. Define the $e$-th epoch as

$$\mathcal{T}_e = \left\{t \mid 2^{e-1} \leq \sum_{\tau=1}^{t} m_\tau < 2^e\right\}. \qquad (17)$$

which is a set of consecutive rounds when the sum of delays is within a given interval. During the $e$-th epoch, the EXP3 algorithm using our doubling trick uses step size $\eta_e = \sqrt{\frac{\ln K}{2^e}}$. Feedback

---

**Algorithm 2** Adaptive EXP3 with delays for unknown $T$ and $\sum_{t=1}^{T} d_t$

---

**Initialization:** Set $\tilde{L}_1^{(i)} = 0$ and $p_1^{(i)} = \frac{1}{K}$ for $i = 1, ..., K$. Set the epoch index $e = 0$ and $\eta_0 = 1$.
**For** $t = 1, ..., T$ **do**

1. Choose an arm $a_t$ at random according to the distribution $\boldsymbol{p_t}$.

2. Obtain a set of delayed costs $l_s^{(a_s)}$ for all $s \in \mathcal{S}_t$, where $a_s$ is the arm played at round $s$.

3. Update the number of missing samples so far

$$m_t = t - \sum_{\tau=1}^{t} |\mathcal{S}_\tau|. \tag{19}$$

4. If $\sum_{\tau=1}^{t} m_\tau \geq 2^e$, then update $e = e + 1$ and initialize $\tilde{L}_t^{(i)} = 0$ for $i = 1, ..., K$.

5. Update the weights of arm $a_s$ for all $s \in \mathcal{S}_t$ such that $s \in \mathcal{T}_e$ using step size $\eta_e = \sqrt{\frac{\ln K}{2^e}}$:

$$\tilde{L}_t^{(a_s)} = \tilde{L}_{t-1}^{(a_s)} + \eta_e \frac{l_s^{(a_s)}}{p_s^{(a_s)}}. \tag{20}$$

6. Update the mixed strategy

$$p_{t+1}^{(i)} = \frac{e^{-\tilde{L}_t^{(i)}}}{\sum_{j=1}^{n} e^{-\tilde{L}_t^{(j)}}}. \tag{21}$$

**End**

---

samples originated in previous epoch are discarded once received. The resulting algorithm is detailed in Algorithm 2.

The next Theorem shows that thanks to our novel doubling trick, Algorithm 2 achieves the same regret guarantee (up to a constant) as in Theorem 1, despite the fact that $T$ **and** $\sum_{t=1}^{T} d_t$ are unknown. We conjecture that the $K^2$ factor replacing $K$ can be improved with a more careful analysis. However, this factor has no effect on the order of the regret when the average delay is larger than $K^2$.

**Theorem 2.** *Let* $\left\{ l_t^{(i)} \right\}$ *be a cost sequence such that* $l_t^{(i)} \in [0, 1]$ *for every* $t, i$. *Let* $\{d_t\}$ *be a delay sequence such that the cost from round* $t$ *is received at round* $t + d_t$. *If player uses Algorithm 2 then*

$$E^{\boldsymbol{a}}\left(R\left(T\right)\right) = E\left\{ \sum_{t=1}^{T} l_t^{(a_t)} - \min_i \sum_{t=1}^{T} l_t^{(i)} \right\} \leq$$

$$O\left( \sqrt{\ln K \left( KT + \sum_{t=1}^{T} \min\{d_t, T - t + 1\} \right)} \right) \leq O\left( \sqrt{\ln K \left( K^2 T + \sum_{t=1}^{T} d_t \right)} \right). \tag{18}$$

*Proof.* See Appendix. $\square$

## 3 Two Player Zero-Sum Game with Delayed Bandit Feedback

In this section we consider a two player zero-sum game where both players play according to the EXP3 algorithm with feedback delays. It is well known that without delays, an algorithm with sublinear regret such as EXP3, played against itself, will converge to a NE (in the ergodic average sense) [18]. Our main result in this section, given in Theorem 3, generalizes this statement for the case of arbitrarily (i.e., adversarially) delayed feedback, and reveals that with delays, convergence to a NE can occur even without sublinear regret.

Let $U$ be the cost matrix, such that when the row player plays $i$ and the column player plays $j$, the first pays a cost of $U(i,j)$ and the second gains a reward of $U(i,j)$ (i.e., a cost of $-U(i,j)$). We assume without loss of generality that $0 \leq U(i,j) \leq 1$ for any $i,j$. Note that if $\boldsymbol{p}_t, \boldsymbol{q}_t \in \Delta^K$ are mixed strategies, then we use the convention that

$$U(\boldsymbol{p}_t, j) \triangleq \sum_{i=1}^{K} p_t^{(i)} U(i,j) \tag{22}$$

and

$$U(\boldsymbol{p}_t, \boldsymbol{q}_t) \triangleq \sum_{i=1}^{K} \sum_{j=1}^{K} p_t^{(i)} q_t^{(j)} U(i,j). \tag{23}$$

Nash Equilibrium (NE) is a key concept in game theory for predicting the outcome of a game. A NE is a strategy profile $(\boldsymbol{p}_t^*, \boldsymbol{q}_t^*)$ such that no player wants to switch a strategy given that the other player keeps his strategy. For our result, we need to define the set of all approximate (and pure) NE:

**Definition 2.** The set of all $\varepsilon$-NE points is

$$\mathcal{N}_\varepsilon = \left\{ (\boldsymbol{p}^*, \boldsymbol{q}^*) \mid U(\boldsymbol{p}^*, \boldsymbol{q}^*) \leq \min_{\boldsymbol{p}} U(\boldsymbol{p}, \boldsymbol{q}) + \varepsilon , \ U(\boldsymbol{p}^*, \boldsymbol{q}^*) \geq \max_{\boldsymbol{q}} U(\boldsymbol{p}, \boldsymbol{q}) - \varepsilon \right\} \tag{24}$$

and the set of NE points is $\mathcal{N}_0$.

The entity that converges to the set of NE in our case is the ergodic average of $(\boldsymbol{p}_t, \boldsymbol{q}_t)$. For the special case of $\eta_\tau = \frac{1}{t}$, the ergodic average of $\boldsymbol{p}_t$ is simply the running average of the sequence $\boldsymbol{p}_t$.

**Definition 3.** The ergodic average of a sequence of distributions $\boldsymbol{p}_t$ is defined as:

$$\bar{\boldsymbol{p}}_t \triangleq \frac{\sum_{\tau=1}^{t} \eta_\tau \boldsymbol{p}_\tau}{\sum_{\tau=1}^{t} \eta_\tau}. \tag{25}$$

We say that $(\bar{\boldsymbol{p}}_T, \bar{\boldsymbol{q}}_T)$ converges in $L^1$ to the set of NE if

$$\lim_{T \to \infty} \underset{(\boldsymbol{p}_T^*, \boldsymbol{q}_T^*) \in \mathcal{N}_0}{\arg\min} E\left\{ \|(\bar{\boldsymbol{p}}_T, \bar{\boldsymbol{q}}_T) - (\boldsymbol{p}_T^*, \boldsymbol{q}_T^*)\|_1 \right\} = 0 \tag{26}$$

which also implies that for every $\varepsilon > 0$

$$\lim_{T \to \infty} \underset{(\boldsymbol{p}_T^*, \boldsymbol{q}_T^*) \in \mathcal{N}_0}{\arg\min} P\left( \|(\bar{\boldsymbol{p}}_T, \bar{\boldsymbol{q}}_T) - (\boldsymbol{p}_T^*, \boldsymbol{q}_T^*)\|_1 \geq \varepsilon \right) = 0. \tag{27}$$

Our theorem below establishes the convergence of EXP3 versus itself to a NE, even under significant delays. Note that the convergence of the ergodic mean to the set of NE is in the $L^1$ sense (so also in probability), which is much stronger than convergence of the expected ergodic mean.

**Theorem 3.** *Let two players play a zero-sum game with a cost matrix $U$ such that $0 \leq U(i,j) \leq 1$ for each $i,j$, using EXP3. The step size sequence of both players is $\{\eta_t\}_{t=1}^{\infty}$. Let the delay sequences of the row player and the column player be $\{d_t^r\}, \{d_t^c\}$, respectively. Let the mixed strategies of the row and column players at round $t$ be $\boldsymbol{p}_t$ and $\boldsymbol{q}_t$, respectively. If*

1. *$\sum_{t=1}^{\infty} \eta_t = \infty$.*

2. *$\lim_{t \to \infty} \eta_t d_t^r < \infty$ and $\lim_{t \to \infty} \eta_t d_t^c < \infty$.*

3. *$\sum_{t=1}^{\infty} d_t^r \eta_t^2 < \infty$ and $\sum_{t=1}^{\infty} d_t^c \eta_t^2 < \infty$.*

*Then, as $T \to \infty$:*

1. *$\left( \frac{\sum_{t=1}^{T} \eta_t \boldsymbol{p}_t}{\sum_{t=1}^{T} \eta_t}, \frac{\sum_{t=1}^{T} \eta_t \boldsymbol{q}_t}{\sum_{t=1}^{T} \eta_t} \right)$ converges in $L^1$ to the set of NE of the zero-sum game.*

2. *$U\left( \frac{\sum_{t=1}^{T} \eta_t \boldsymbol{p}_t}{\sum_{t=1}^{T} \eta_t}, \frac{\sum_{t=1}^{T} \eta_t \boldsymbol{q}_t}{\sum_{t=1}^{T} \eta_t} \right)$ converges in $L^1$ to $\min_{\boldsymbol{p}} \max_{j} U(\boldsymbol{p}, j) = \max_{\boldsymbol{q}} \min_{i} U(i, \boldsymbol{q})$, which is the value of the game.*

Somewhat surprisingly, the delays do not have to be bounded (in $t$) for the convergence to NE to hold. Key examples of application of Theorem 3 are:

- For bounded delays $d_t^r \leq D$ and $d_t^c \leq D$ for all $t$:
  - For a finite horizon $T$ one can choose $\eta_t = \frac{1}{\sqrt{T}}$ for all $t$.
  - For the infinite horizon case one can choose any $\eta_t$ such that $\sum_{t=1}^{\infty} \eta_t^2 < \infty$ and $\sum_{t=1}^{\infty} \eta_t = \infty$.
- For unbounded sublinear delays such as $d_t^r \leq \sqrt{t}$ and $d_t^c \leq \sqrt{t}$ for all $t$, one can choose $\eta_t = \frac{1}{t^{2/3}}$.
- For unbounded superlinear delays such as $d_t^r \leq t \log t$ and $d_t^c \leq t \log t$, one can choose $\eta_t = \frac{1}{t(\log t)(\log \log t)}$.

In general, the feedback of the players does not need to be synchronized, and they may have a completely different sequence of delays.

Next we show that the ergodic average of the EXP3 strategies converges to the set of NE even in a delayed feedback scenario where EXP3 has linear regret, so the "no-regret" property does not hold.

**Proposition 1.** *Let the mixed strategies of the row and column players at round $t$ be $\boldsymbol{p}_t$ and $\boldsymbol{q}_t$, respectively. There exist $\{d_t^r, d_t^c\}_t$ and a cost sequence $\left\{l_t^{(1)}, ..., l_t^{(K)}\right\}_t$ such that*

$$E^{\boldsymbol{a}} \left\{ \sum_{t=1}^{T} l_t^{(a_t)} - \min_{\boldsymbol{p}} \sum_{t=1}^{T} p^{(i)} l_t^{(i)} \right\} \geq \left(1 - \frac{1}{K}\right) \frac{T}{2} \qquad (28)$$

*but still the step sizes $\{\eta_t\}$ for Algorithm 1 can be chosen such that the conclusion of Theorem 3 still holds ("convergence to NE").*

*Proof.* Let $d_t^r = d_t^c = d_t = t$ and $\eta_t = \frac{1}{t \log t}$ for all $t$, for which $d_t \eta_t^2 = \frac{1}{t \log^2 t}$ so $\sum_{t=1}^{T} \eta_t = \infty$, $\sum_{t=1}^{T} \eta_t^2 < \infty$, $\sum_{t=1}^{T} d_t \eta_t^2 < \infty$ and $\lim_{t \to \infty} \eta_t d_t = 0$. Hence, Theorem 3 applies and $(\bar{\boldsymbol{p}}_T, \bar{\boldsymbol{q}}_T)$ converges in $L^1$ to the set of NE of the game. However, the feedback for the last $\frac{T}{2}$ rounds is never received. Therefore, the mixed strategies $\boldsymbol{p}_t$ and $\boldsymbol{q}_t$ stay constant for all $t \geq \frac{T}{2}$. Consider the sequence of costs $l_t^{(i)} = 0$ for all $i$ and all $t \leq \frac{T}{2}$ and $l_t^{(1)} = 0$, $l_t^{(j)} = 1$ for all $j > 1$ and all $t > \frac{T}{2}$. This sequence yields an expected regret of exactly $\left(1 - \frac{1}{K}\right) \frac{T}{2}$. $\qquad \square$

## 4 Conclusions

In this paper, we analyzed the regret of the EXP3 algorithm subjected to an arbitrary (i.e., adversarial) sequence $d_t$ of feedback delays. We have shown that the expected regret is $O\left(\sqrt{\ln K \left(KT + \sum_{t=1}^{T} d_t\right)}\right)$. This shows that the EXP3 algorithm is inherently robust to delays, since for $d_t \leq K$ the order of magnitude of the regret does not change (as a function of $T$ and $K$) from the famous $O\left(\sqrt{K \ln KT}\right)$. We have also proved that the convergence of the ergodic average to a Nash equilibrium under delays is a more robust property than the no-regret property of EXP3. The ergodic average converges to the set of Nash equilibria even under super-linear delays where EXP3 has a linear regret in $T$. This serves as a concrete example where competing versus another agent is essentially easier than competing versus an omnipotent adversary, even if the other agent is not subject to any delays.

## Acknowledgments

This research was supported by the Koret Foundation grant for Smart Cities and Digital Living. Zhengyuan Zhou gratefully acknowledges IBM Goldstine Fellowship. Xi Chen is supported by NSF via IIS-1845444.

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
