[Supplementary Material · Appendix - Online EXP3 Learning in Adversarial Bandits with Delayed Feedback.pdf]

# 5 Supplementary Material

## 5.1 Auxiliary Lemmas

The next lemma shows smoothness properties of the softmax function.

**Lemma 2.** *Let* $h_i(\boldsymbol{x}) = \frac{e^{-x_i}}{\sum_{j=1}^{K} e^{-x_j}}$ *and* $h(\boldsymbol{x}) = (h_1(\boldsymbol{x}), ..., h_K(\boldsymbol{x}))$. *Then for any* $\boldsymbol{x} \in \mathbb{R}^K$ *and* $\Delta \in \mathbb{R}_+^K$

$$\|h(\boldsymbol{x}) - h(\boldsymbol{x} + \Delta)\|_1 \leq 2 \langle h(\boldsymbol{x}), \Delta \rangle. \tag{29}$$

*Proof.* For all $\boldsymbol{x} \in \mathbb{R}^K$ and $\Delta \in \mathbb{R}_+^K$

$$h_i(\boldsymbol{x} + \Delta) - h_i(\boldsymbol{x}) = \frac{e^{-x_i - \Delta_i}}{\sum_{j=1}^{K} e^{-x_j - \Delta_j}} - \frac{e^{-x_i}}{\sum_{j=1}^{K} e^{-x_j}} \underset{(a)}{\geq}$$

$$\frac{e^{-x_i - \Delta_i} - e^{-x_i}}{\sum_{j=1}^{K} e^{-x_j}} = \left(e^{-\Delta_i} - 1\right) h(\boldsymbol{x}) \underset{(b)}{\geq} -\Delta_i h_i(\boldsymbol{x}) \tag{30}$$

where (a) follows since $\sum_{j=1}^{K} e^{-x_j - \Delta_j} \leq \sum_{j=1}^{K} e^{-x_j}$ and (b) since $1 - x \leq e^{-x}$ for all $x \geq 0$.

We also have for all $\boldsymbol{x} \in \mathbb{R}^K$ and $\Delta \in \mathbb{R}_+^K$ that

$$h_i(\boldsymbol{x} + \Delta) - h_i(\boldsymbol{x}) = \frac{e^{-x_i - \Delta_i}}{\sum_{j=1}^{K} e^{-x_j - \Delta_j}} - \frac{e^{-x_i}}{\sum_{j=1}^{K} e^{-x_j}} \underset{(a)}{\leq}$$

$$\frac{e^{-x_i - \Delta_i}}{\sum_{j=1}^{K} e^{-x_j - \Delta_j}} - \frac{e^{-x_i - \Delta_i}}{\sum_{j=1}^{K} e^{-x_j}} = h_i(\boldsymbol{x} + \Delta) \left(1 - \frac{\sum_{j=1}^{K} e^{-x_j - \Delta_j}}{\sum_{l=1}^{K} e^{-x_l}}\right) =$$

$$h_i(\boldsymbol{x} + \Delta) \frac{\sum_{j=1}^{K} e^{-x_j} \left(1 - e^{-\Delta_j}\right)}{\sum_{l=1}^{K} e^{-x_l}} \underset{(b)}{\leq} h_i(\boldsymbol{x} + \Delta) \frac{\sum_{j=1}^{K} \Delta_j e^{-x_j}}{\sum_{l=1}^{K} e^{-x_l}} \tag{31}$$

where (a) follows since $e^{-x_j} \geq e^{-x_j - \Delta_j}$ and (b) since $1 - x \leq e^{-x}$ for all $x \geq 0$. Combining the two inequalities we conclude that

$$\|h(\boldsymbol{x}) - h(\boldsymbol{x} + \Delta)\|_1 = \sum_{i=1}^{K} |h_i(\boldsymbol{x}) - h_i(\boldsymbol{x} + \Delta)| \underset{(a)}{\leq}$$

$$\sum_{i=1}^{K} \Delta_i h_i(\boldsymbol{x}) + \sum_{i=1}^{K} h_i(\boldsymbol{x} + \Delta) \left(\sum_{j=1}^{K} \frac{\Delta_j e^{-x_j}}{\sum_{l=1}^{K} e^{-x_l}}\right) =$$

$$\langle h(\boldsymbol{x}), \Delta \rangle + \left(\sum_{j=1}^{K} \left(\Delta_j \frac{e^{-x_j}}{\sum_{l=1}^{K} e^{-x_l}}\right)\right) \sum_{i=1}^{K} h_i(\boldsymbol{x} + \Delta) \underset{(b)}{=} 2 \langle h(\boldsymbol{x}), \Delta \rangle \tag{32}$$

where (a) follows since (30) and (31) show that for all $i$

$$|h_i(\boldsymbol{x} + \Delta) - h_i(\boldsymbol{x})| \leq \max\left\{\Delta_i h_i(\boldsymbol{x}), h_i(\boldsymbol{x} + \Delta) \frac{\sum_{j=1}^{K} \Delta_j e^{-x_j}}{\sum_{j=1}^{K} e^{-x_j}}\right\} \tag{33}$$

and (b) follows since $\sum_{i=1}^{K} h_i(\boldsymbol{x} + \Delta) = 1$ by definition. $\qquad\square$

The next lemma analyzes the contribution of the "no-delay" term to the expected regret.

**Lemma 3.** *Let* $\eta_t$ *be a non-increasing a sequence of step sizes. Let* $\left\{l_t^{(i)}\right\}$ *be a cost sequence such that* $l_t^{(i)} \in [0, 1]$ *for every* $t, i$. *Let* $\{d_t\}$ *be a delay sequence such that the reward from round* $t$ *is received at round* $t + d_t$. *Let* $\mathcal{S}_t$ *be the set of costs (feedback samples) received at round* $t$. *Then*

$$E^{\boldsymbol{a}}\left\{\sum_{t=1}^{T} \sum_{s \in \mathcal{S}_t} \eta_s \left\langle l_s, \boldsymbol{p}_{s_-}\right\rangle - \min_i \sum_{t=1}^{T} \eta_t l_t^{(i)}\right\} \leq \ln K + \frac{K}{2} \sum_{t=1}^{T} \eta_t^2. \tag{34}$$

*Proof.* Define $s_-, s_+$ as the steps a moment before and after using the feedback from round $s$, respectively. These steps are taking place in round $t$ if $s \in \mathcal{S}_t$, and $\boldsymbol{p}_{s_-}$ is the computed probability vector at $s_-$. Define $\Phi(t) = -\ln\left(\sum_{i=1}^{K} e^{-\tilde{L}_t^{(i)}}\right)$ and $\tilde{l}_t = \left(0, ..., \frac{l_t^{(a_t)}}{p_t^{(a_t)}}, ..., 0\right)$. We have

$$\Phi(s_+) - \Phi(s_-) = -\ln\left(\frac{\sum_{i=1}^{K} e^{-\tilde{L}_{s_-}^{(i)}} e^{-\eta_s \tilde{l}_s^{(i)}}}{\sum_{j=1}^{K} e^{-\tilde{L}_{s_-}^{(j)}}}\right) = -\ln\left(\sum_{i=1}^{K} p_{s_-}^{(i)} e^{-\eta_s \tilde{l}_s^{(i)}}\right) \underset{(a)}{\geq}$$

$$-\ln\left(\sum_{i=1}^{K} p_{s_-}^{(i)} \left(1 - \eta_s \tilde{l}_s^{(i)} + \frac{1}{2}\eta_s^2 \left(\tilde{l}_s^{(i)}\right)^2\right)\right) =$$

$$-\ln\left(1 - \sum_{i=1}^{K} p_{s_-}^{(i)} \left(\eta_s \tilde{l}_s^{(i)} - \frac{1}{2}\eta_s^2 \left(\tilde{l}_s^{(i)}\right)^2\right)\right) \underset{(b)}{\geq} \eta_s \sum_{i=1}^{K} p_{s_-}^{(i)} \tilde{l}_s^{(i)} - \frac{\eta_s^2}{2}\sum_{i=1}^{K} p_{s_-}^{(i)} \left(\tilde{l}_s^{(i)}\right)^2 \quad (35)$$

where (a) follows since $e^{-x} \leq 1 - x + \frac{1}{2}x^2$ and (b) since $\ln(1-x) \leq -x$. Taking the expectation on both sides of (35) yields

$$E^{\boldsymbol{a}}\{\Phi(s_+) - \Phi(s_-)\} \geq E^{\boldsymbol{a}}\left\{\eta_s \sum_{i=1}^{K} p_{s_-}^{(i)} \tilde{l}_s^{(i)} - \frac{\eta_s^2}{2}\sum_{i=1}^{K} p_{s_-}^{(i)} \left(\tilde{l}_s^{(i)}\right)^2\right\} \underset{(a)}{=}$$

$$E^{\boldsymbol{a}}\left\{\eta_s \sum_{i=1}^{K} p_{s_-}^{(i)} E^{\boldsymbol{a}}\left\{\tilde{l}_s^{(i)} \mid \mathcal{F}_{s_-}\right\}\right\} - \frac{\eta_s^2}{2} E^{\boldsymbol{a}}\left\{\sum_{i=1}^{K} p_{s_-}^{(i)} E^{\boldsymbol{a}}\left\{\left(\tilde{l}_s^{(i)}\right)^2 \mid \mathcal{F}_{s_-}\right\}\right\} \underset{(b)}{=}$$

$$E^{\boldsymbol{a}}\left\{\eta_s \left\langle \boldsymbol{l}_s, \boldsymbol{p}_{s_-}\right\rangle\right\} - \frac{\eta_s^2}{2}\sum_{i=1}^{K} \left(l_s^{(i)}\right)^2 \geq E^{\boldsymbol{a}}\left\{\eta_s \left\langle \boldsymbol{l}_s, \boldsymbol{p}_{s_-}\right\rangle\right\} - \frac{\eta_s^2}{2}K \quad (36)$$

where (a) uses $p_{s_-}^{(i)} \in \mathcal{F}_{s_-}$ and (b) uses $p_s^{(i)} \in \mathcal{F}_{s_-}$ (since $s < s_-$) together with the fact that $\tilde{l}_s^{(i)}$ is $\frac{l_s^{(i)}}{p_s^{(i)}}$ with probability $p_s^{(i)}$ and zero otherwise. Note that $a_s$ is independent of $\mathcal{F}_{s_-}$ since by definition the feedback from $a_s$ was not received until round $s_-$. Hence, by iterating (36) over $s$ we obtain

$$E^{\boldsymbol{a}}\left\{\Phi\left(s_T^+\right) - \Phi(1)\right\} = E^{\boldsymbol{a}}\left\{\sum_{t=1}^{T} \sum_{s \in \mathcal{S}_t} (\Phi(s_+) - \Phi(s_-))\right\} \geq$$

$$E^{\boldsymbol{a}}\left\{\sum_{t=1}^{T} \sum_{s \in \mathcal{S}_t} \eta_s \left\langle \boldsymbol{l}_s, \boldsymbol{p}_{s_-}\right\rangle\right\} - \frac{K}{2}\sum_{t=1}^{T} \eta_t^2. \quad (37)$$

where $s_T \in \mathcal{S}_T$ is the last feedback to be updated at round $T$. Now we upper bound $\Phi\left(s_T^+\right) - \Phi(1)$. We have for every $i$

$$E^{\boldsymbol{a}}\left\{\Phi\left(s_T^+\right) - \Phi(1)\right\} = -E^{\boldsymbol{a}}\left\{\ln\left(\sum_{j=1}^{K} e^{-\tilde{L}_{s_T^+}^{(j)}}\right) - \ln K\right\} \underset{(a)}{\leq}$$

$$E^{\boldsymbol{a}}\left\{\tilde{L}_{s_T^+}^{(i)} + \ln K\right\} \underset{(b)}{\leq} \sum_{t=1}^{T} \eta_t l_t^{(i)} + \ln K \quad (38)$$

where (a) follows by omitting positive terms from $\sum_{i=1}^{K} e^{-\tilde{L}_t^{(i)}}$ and (b) since we are adding $\eta_t l_t^{(i)}$ (positive) terms of rounds whose feedback was not received before round $T$. Combining (37) and (38), we obtain for all $i = 1, ..., K$

$$E^{\boldsymbol{a}}\left\{\sum_{t=1}^{T} \sum_{s \in \mathcal{S}_t} \eta_s \left\langle \boldsymbol{l}_s, \boldsymbol{p}_{s_-}\right\rangle - \sum_{t=1}^{T} \eta_t l_t^{(i)}\right\} \leq \ln K + \frac{K}{2}\sum_{t=1}^{T} \eta_t^2. \quad (39)$$

$\square$

The next lemma is necessary to analyze the contribution of the "delay term" to the expected regret.

**Lemma 4.** *Let $\{\eta_t\}$ be a non-increasing positive sequence. Let $d_t$ be the delay of the cost of the action at round $t$. Let $\mathcal{S}_r$ be the set of feedback samples received at round $t$, and define $\mathcal{S}_{t,s} = \{r \in \mathcal{S}_t; r < s\}$, which is the set of feedback samples that the algorithm uses before the feedback from round $s$ is used. Define the set $\mathcal{M}$ of all samples that have not been received by round $T$. Then*

$$\sum_{t=1}^{T} \sum_{s \in \mathcal{S}_t} \eta_s \left( \sum_{q \in \mathcal{S}_{t,s}} \eta_q + \sum_{r=s}^{t-1} \sum_{q \in \mathcal{S}_r} \eta_q \right) \leq 2 \sum_{t \notin \mathcal{M}} \eta_t^2 d_t. \tag{40}$$

*Proof.* The quantity $Q_{s,t} \triangleq \sum_{q \in \mathcal{S}_{t,s}} \eta_q + \sum_{r=s}^{t-1} \sum_{q \in \mathcal{S}_r} \eta_q$ is a weighted count of the number of feedback samples received and used between round $s$ and round $t$, before the feedback from round $s$ is used. We want to upper bound $\sum_{t=1}^{T} \sum_{s \in \mathcal{S}_t} \eta_s Q_{s,t}$ for all possible delay sequences $\{d_t\}$. We do so by (over) counting the number of appearances of each feedback from the $T$ feedback samples, in the different $Q_{s,t}$ "buckets". There are two possible cases of feedback samples being counted, so we write $Q_{s,t} = Q_{s,t}^1 + Q_{s,t}^2$.

- A feedback from $q \geq s$ is received and used before $s$ is used: there are a maximum of $d_s$ feedback samples of this type that can each contribute $\eta_q \leq \eta_s$ with $q \geq s$ to $Q_{s,t}^1$ for $s \in \mathcal{S}_t$ (since $\eta_t$ is non-increasing). We over count them by giving each $Q_{s,t}^1$ term all of its $d_s$ possible samples of this type. So

$$\sum_{t=1}^{T} \sum_{s \in \mathcal{S}_t} \eta_s Q_{s,t}^1 \leq \sum_{t=1}^{T} \sum_{s \in \mathcal{S}_t} \eta_s^2 d_s = \sum_{t \notin \mathcal{M}} \eta_t^2 d_t. \tag{41}$$

- A feedback from $q < s$ is received and used before $s$ is used: the samples from round $q$ can contribute to a maximum of $d_q$ different $Q_{s,t}^2$ terms, all with $s \geq q$. This follows simply because the feedback from $q$ is not received before $q + d_q$. Denote by $\Gamma_q$ the set of rounds $s$ such that the samples from round $q$ contribute to $Q_{s,t}^2$. Then

$$\sum_{t=1}^{T} \sum_{s \in \mathcal{S}_t} \eta_s Q_{s,t}^2 \underset{(a)}{=} \sum_{q \notin \mathcal{M}} \sum_{s \in \Gamma_q} \eta_s \eta_q \underset{(b)}{\leq} \sum_{q \notin \mathcal{M}} \eta_q^2 |\Gamma_q| \leq \sum_{q \notin \mathcal{M}} \eta_q^2 d_q \tag{42}$$

where (a) follows since only rounds $q$ whose feedback is received sometime before $T$ are counted in $Q_{s,t}^2$ for some $s,t$. Inequality (b) uses $\eta_s^2 \leq \eta_q^2$ since $\eta_t$ is non-increasing and $s \geq q$ for all $s \in \Gamma_q$.

Adding (41) and (42) we obtain (40). $\qquad \square$

## 5.2   Proof of Theorem 2

*Proof.* Define $\mathcal{M}_e$ as the set of feedback samples for costs in epoch $e$ that are not received within epoch $e$. Denote by $T_e = \max \mathcal{T}_e$ the last round in $\mathcal{T}_e$. Note that $\mathcal{T}_e$ is the set of consecutive rounds from $T_{e-1}+1$ to $T_e$. Every round $t \in \mathcal{T}_e$ such that $t \notin \mathcal{M}_e$ contributes exactly $d_t$ to $\sum_{\tau=T_{e-1}+1}^{T_e} m_\tau$, since the $t$-th feedback is missing for $d_t$ rounds some time between $T_{e-1} + 1$ and $T_e$. Therefore

$$\sum_{t \in \mathcal{T}_e, t \notin \mathcal{M}_e} d_t \leq \sum_{\tau=T_{e-1}+1}^{T_e} m_\tau \underset{(a)}{\leq} 2^{e-1} \tag{43}$$

where (a) uses that if $\sum_{\tau=T_{e-1}+1}^{T_e} m_\tau > 2^{e-1}$ then $\sum_{\tau=1}^{T_e} m_\tau \geq 2^{e-1} + 2^{e-1} = 2^e$ so epoch $e + 1$ should have been already started. We apply Theorem 1 separately on every epoch, which yields

$$R_e \triangleq E^{\boldsymbol{a}} \left\{ \sum_{t \in \mathcal{T}_e} l_t^{(a_t)} - \min_i \sum_{t \in \mathcal{T}_e} l_t^{(i)} \right\} \leq \frac{\ln K}{\eta_e} + \eta_e \left( \frac{K}{2} |\mathcal{T}_e| + 4 \sum_{t \in \mathcal{T}_e, t \notin \mathcal{M}_e} d_t \right) + |\mathcal{M}_e|. \tag{44}$$

Now we want to find the maximal $|\mathcal{M}_e|$ such that $\sum_{\tau=T_{e-1}+1}^{T_e} m_\tau \leq 2^{e-1}$ is still possible. The "cheapest" way to increase $|\mathcal{M}_e|$ is when the feedback from round $T_e$ is delayed by one (contributes 1 to $\sum_{\tau=T_{e-1}+1}^{T_e} m_\tau$), the feedback from round $T_e - 1$ is delayed by two (contributes 2 to $\sum_{\tau=T_{e-1}+1}^{T_e} m_\tau$) and so on, which gives

$$\sum_{i=1}^{|\mathcal{M}_e|} i = \frac{|\mathcal{M}_e|\,(|\mathcal{M}_e|+1)}{2} \leq 2^{e-1} \implies |\mathcal{M}_e| \leq 2^{\frac{e}{2}} \tag{45}$$

so by choosing $\eta_e = \sqrt{\frac{\ln K}{2^e}}$ we obtain

$$R_e \leq \sqrt{\ln K}\left(2^{\frac{e}{2}} + 2^{-\frac{e}{2}}\left(\frac{K}{2}|\mathcal{T}_e| + 4\sum_{t\in\mathcal{T}_e, t\notin\mathcal{M}_e} d_t\right)\right) + 2^{\frac{e}{2}} \underset{(a)}{\leq}$$
$$3\cdot 2^{\frac{e}{2}}\sqrt{\ln K} + 2^{-\frac{e}{2}-1}|\mathcal{T}_e|\,K\sqrt{\ln K} + 2^{\frac{e}{2}} \tag{46}$$

where (a) follows from (43). Denote the last epoch by $E$. Hence, we conclude that

$$E^a\{R(T)\} = \sum_{e=1}^{E} R_e \leq \left(3\sqrt{\ln K}+1\right)\sum_{e=1}^{E} 2^{\frac{e}{2}} + \frac{K}{2}\sqrt{\ln K}\sum_{e=1}^{E}|\mathcal{T}_e|\,2^{-\frac{e}{2}} \leq$$
$$\sqrt{2}\left(3\sqrt{\ln K}+1\right)\frac{2^{\frac{E}{2}}-1}{\sqrt{2}-1} + \frac{K}{2}\sqrt{\ln K}\sum_{e=1}^{E}|\mathcal{T}_e|\,2^{-\frac{e}{2}} \underset{(a)}{\leq}$$
$$15\left(\sqrt{\ln K}+1\right)\sqrt{\sum_{t=1}^{T} d_t} + \frac{5}{2}K\sqrt{T\ln K} = O\left(\sqrt{\ln K\left(K^2 T + \sum_{t=1}^{T} d_t\right)}\right) \tag{47}$$

where in (a) we used that

$$\sum_{t=1}^{T} d_t \geq \sum_{t=1}^{T}\min\{d_t, T-t+1\} = \sum_{t=1}^{T} m_t \geq \sum_{t=1}^{T_E} m_t \geq 2^{E-1} \tag{48}$$

and also that $\sum_{e=1}^{E}|\mathcal{T}_e|\,2^{-\frac{e}{2}}$ subject to $\sum_{e=1}^{E}|\mathcal{T}_e| = T$ is maximized when there are only $\lceil\log_2 T\rceil$ epochs with length $2^e$ to epoch $e$ (maximal length possible), so

$$\sum_{e=1}^{E}|\mathcal{T}_e|\,2^{-\frac{e}{2}} \leq \sum_{e=1}^{\lceil\log_2 T\rceil} 2^{\frac{e}{2}} \leq \sqrt{2}\frac{2^{\frac{\lceil\log_2 T\rceil}{2}}-1}{\sqrt{2}-1} \leq 5\sqrt{T} \tag{49}$$

$\square$

## 5.3 Proof of Theorem 3

*Proof.* We need to show that $(\bar{\boldsymbol{p}}_T, \bar{\boldsymbol{q}}_T)$ converges in $L^1$ to the set of NE of the game as $T \to \infty$. Let $\varepsilon > 0$. Define the ergodic average of the value of the game by

$$\overline{U}_T = \frac{\sum_{t=1}^{T} \eta_t U(\boldsymbol{p}_t, \boldsymbol{q}_t)}{\sum_{t=1}^{T} \eta_t}. \tag{50}$$

By using EXP3 with cost sequence $l_{r,t}^{(i)} = U(i, \boldsymbol{q}_t)$ we know from Lemma 1 that the row player guarantees that for any column strategy, in particular $\boldsymbol{q}_t$, and any row strategy $\boldsymbol{p}$, possibly random, we have

$$E^a\left\{\sum_{t=1}^{T} \eta_t\left(U(\boldsymbol{p}_t, \boldsymbol{q}_t) - U(\boldsymbol{p}, \boldsymbol{q}_t)\right)\right\} \leq \ln K + \frac{K}{2}\sum_{t=1}^{T} \eta_t^2 + 4\sum_{t=1}^{T} \eta_t^2 d_t^r + \sum_{t\in\mathcal{M}^r} \eta_t \tag{51}$$

where the set of missing samples is $\mathcal{M}^r = \{t \,|\, t + d_t^r > T\}$. Define $t^*(T) = \min \mathcal{M}^r$, and note that $t^*(T) \to \infty$ as $T \to \infty$ since $t + d_t^r \geq t$, and $f(t) = t$ is increasing. Since $\eta_t$ is non-increasing then

$$\sum_{t \in \mathcal{M}^r} \eta_t \leq |\mathcal{M}^r| \eta_{t^*(T)} \leq (T - t^*(T) + 1) \eta_{t^*(T)} \leq d_{t^*(T)}^r \eta_{t^*(T)}. \qquad (52)$$

Therefore there exists a $T_1 > 0$ such that for all $T > T_1$

$$E^{\boldsymbol{a}} \left\{ \overline{U}_T - U(\boldsymbol{p}, \bar{\boldsymbol{q}}_T) \right\} = E^{\boldsymbol{a}} \left\{ \frac{\sum_{t=1}^{T} \eta_t \left( U(\boldsymbol{p}_t, \boldsymbol{q}_t) - U(\boldsymbol{p}, \boldsymbol{q}_t) \right)}{\sum_{t=1}^{T} \eta_t} \right\} \underset{(a)}{\leq}$$

$$\frac{d_{t^*(T)}^r \eta_{t^*(T)} + \ln K + \frac{K}{2} \sum_{t=1}^{T} \eta_t^2 + 4 \sum_{t=1}^{T} \eta_t^2 d_t^r}{\sum_{t=1}^{T} \eta_t} \underset{(b)}{\leq} \frac{\varepsilon}{2} \qquad (53)$$

where (a) is (51) and (b) follows since $d_t^r \eta_t \to 0$ as $t \to \infty$, $\sum_{t=1}^{\infty} \eta_t = \infty$ and $\sum_{t=1}^{\infty} d_t^r \eta_t^2 < \infty$. By also using EXP3 with cost sequence $l_{c,t}^{(j)} = 1 - U(\boldsymbol{p}_t, j)$, we know from Lemma 1 that the column player guarantees that for any row strategy, in particular $\boldsymbol{p}_t$ and any column strategy $\boldsymbol{q}$, possibly random, we have

$$E^{\boldsymbol{a}} \left\{ \sum_{t=1}^{T} \eta_t \left( U(\boldsymbol{p}_t, \boldsymbol{q}) - U(\boldsymbol{p}_t, \boldsymbol{q}_t) \right) \right\} \leq \ln K + \frac{K}{2} \sum_{t=1}^{T} \eta_t^2 + 4 \sum_{t=1}^{T} \eta_t^2 d_t^c + \sum_{t \in \mathcal{M}^c} \eta_t. \qquad (54)$$

Therefore there exists a $T_2 > 0$ such that for all $T > T_2$

$$E^{\boldsymbol{a}} \left\{ U(\bar{\boldsymbol{p}}_T, \boldsymbol{q}) - \overline{U}_T \right\} = \frac{E^{\boldsymbol{a}} \left\{ \sum_{t=1}^{T} \eta_t \left( U(\boldsymbol{p}_t, \boldsymbol{q}) - U(\boldsymbol{p}_t, \boldsymbol{q}_t) \right) \right\}}{\sum_{t=1}^{T} \eta_t} \underset{(a)}{\leq}$$

$$\frac{d_{t^*(T)}^c \eta_{t^*(T)} + \ln K + \frac{K}{2} \sum_{t=1}^{T} \eta_t^2 + 4 \sum_{t=1}^{T} \eta_t^2 d_t^c}{\sum_{t=1}^{T} \eta_t} \underset{(b)}{\leq} \frac{\varepsilon}{2} \qquad (55)$$

where (a) is (54) and (b) follows since $d_t^c \eta_t \to 0$ as $t \to \infty$, $\sum_{t=1}^{\infty} \eta_t = \infty$ and $\sum_{t=1}^{\infty} d_t^c \eta_t^2 < \infty$.

Now, define $\boldsymbol{p}_T^b$ as the best-response to $\bar{\boldsymbol{q}}_T$, which is a random vector that is a function of the random vector $\bar{\boldsymbol{q}}_T$

$$\boldsymbol{p}_T^b = \arg\min_{\boldsymbol{p}'} U(\boldsymbol{p}', \bar{\boldsymbol{q}}_T) \qquad (56)$$

together with $\boldsymbol{q}_T^b$, the best-response to $\bar{\boldsymbol{p}}_T$, which is a random vector that is a function of the random vector $\bar{\boldsymbol{p}}_T$:

$$\boldsymbol{q}_T^b = \arg\max_{\boldsymbol{q}'} U(\bar{\boldsymbol{p}}_T, \boldsymbol{q}'). \qquad (57)$$

Hence, by choosing $\boldsymbol{p} = \boldsymbol{p}_T^b$, $\boldsymbol{q} = \bar{\boldsymbol{q}}_T$ in (53) and (55) and adding them together we conclude that for all $T > \max\{T_1, T_2\}$

$$E^{\boldsymbol{a}} \left\{ \left| U(\bar{\boldsymbol{p}}_T, \bar{\boldsymbol{q}}_T) - \min_{\boldsymbol{p}'} U(\boldsymbol{p}', \bar{\boldsymbol{q}}_T) \right| \right\} \underset{(a)}{=} E^{\boldsymbol{a}} \left\{ \overline{U}_T - U(\boldsymbol{p}_T^b, \bar{\boldsymbol{q}}_T) \right\} + E^{\boldsymbol{a}} \left\{ U(\bar{\boldsymbol{p}}_T, \bar{\boldsymbol{q}}_T) - \overline{U}_T \right\} \leq \varepsilon \qquad (58)$$

where (a) follows since $U(\bar{\boldsymbol{p}}_T, \bar{\boldsymbol{q}}_T) \geq U(\boldsymbol{p}_T^b, \bar{\boldsymbol{q}}_T)$. By choosing instead $\boldsymbol{p} = \bar{\boldsymbol{p}}_T$, $\boldsymbol{q} = \boldsymbol{q}_T^b$ in (53) and (55) and adding them together we conclude that for all $T > \max\{T_1, T_2\}$

$$E^{\boldsymbol{a}} \left\{ \left| U(\bar{\boldsymbol{p}}_T, \bar{\boldsymbol{q}}_T) - \max_{\boldsymbol{q}'} U(\bar{\boldsymbol{p}}_T, \boldsymbol{q}') \right| \right\} \underset{(a)}{=} E^{\boldsymbol{a}} \left\{ \overline{U}_T - U(\bar{\boldsymbol{p}}_T, \bar{\boldsymbol{q}}_T) \right\} + E^{\boldsymbol{a}} \left\{ U(\bar{\boldsymbol{p}}_T, \boldsymbol{q}_T^b) - \overline{U}_T \right\} \leq \varepsilon \qquad (59)$$

where (a) follows since $U(\bar{\boldsymbol{p}}_T, \bar{\boldsymbol{q}}_T) \leq U(\bar{\boldsymbol{p}}_T, \boldsymbol{q}_T^b)$. Equations (58) and (59) show that $(\bar{\boldsymbol{p}}_T, \bar{\boldsymbol{q}}_T)$ is in $\mathcal{N}_\varepsilon$ in the $L^1$ sense. Since $\varepsilon > 0$ is arbitrary, and $\mathcal{N}_\varepsilon$ is monotonically decreasing to $\mathcal{N}_0$ as $\varepsilon \to 0$, we conclude that $(\bar{\boldsymbol{p}}_T, \bar{\boldsymbol{q}}_T)$ converges in $L^1$ to $\mathcal{N}_0$, which is the set of NE of the game. By Markov's inequality, it follows that $(\bar{\boldsymbol{p}}_T, \bar{\boldsymbol{q}}_T)$ converges in probability to the set of NE. Since $U$ is linear, $U(\bar{\boldsymbol{p}}_T, \bar{\boldsymbol{q}}_T)$ converges to the value of the game $\qquad \square$