[Reviews · NeurIPS 2019]

Reviewer 1



i. Novelty: The paper addressed the similar problem setups introduced in [1], [3], [10], [11] in an adversarial bandit feedback setup and showed that the classical EXP3 algorithm is robust enough to be applied in the delayed feedback setup as well. However I have a major issue with their proposed algorithms which seems erroneous due to the choice of the learning rate (\eta) which requires the knowledge of the delays (d_t) -- but according to the problem formulation this is unknown to the learner. Then the whole technique seems to be pointless! ii: Choice of learning rate (\eta): How to select the learning rate \eta? The optimal learning rate seem to depend on the delays (d_t), e.g. Thm 1, Line-146 etc., but those are unknown to the learner. The claims of the paper stands vacuous if the proposed technique requires the knowledge of delays, where lies the major challenge of the problem addressed. iii. Writing: Overall, the paper is well structured and as easy to follow. However there are some minor issues which could be taken care of: a) In Eqn. (1) and (4), it is better and least confusing to use just \min_i \sum_{t}l_t^{(i)}. b) Line 91: Unexpected occurrence of "," c) How is p_1 initialized in Alg-1 and 2 (uniformly I presume)? d) Introducing the notation \lamda^* in Thm 2 is unnecessary e) \eta_e expression incorrect in Eqn (9). etc. iv. Clarity of the results and comparison with earlier works: a) Overall, the paper does not draw a clear picture of how their result improves over the existing work, which would have been very useful to understand their precise contributions. b) Some intuition behind ergodic average (Eqn. 15) would be appreciated. c) Dependence of delay is somewhat unclear in the second setting (of two player game). In my understanding the statement "the delays do not have to be bounded (in t) for the convergence to NE to hold" (Line-186) is tad confusing, e.g. is there a way to set \eta_t, \gamma_t if d_t = e^t, and still achieve "ergodic convergence"? Does Thm-2 yield back the classical NE convergence result of no-regret algorithm [14] when there is no delay, i.e. d_t = 0 \forall t --- explain how? v. Experiments: Its disappointing that the paper does not provide any empirical studies to validate the theoretical guarantees/ comparative performances with existing works.

Reviewer 2



Originality: The statement of the theorems are new and main technical contribution is the analysis of existing algorithm EXP3. Quality: The paper looks technically sound and the statement of the theorems look complete. Clarity: The paper is clearly written. Significance: The paper has very good significance to the field. There are a steam of paper studying delayed feedback in multi armed bandits and this paper makes a very nice contribution. It's ideas and techniques are likely to be applicable to follow up work.

Reviewer 3



Major Comments. While online learning problems with delay are important problems, I have two concerns about the paper. First, there is a lack of comparison with relevant literature, which hinders the readers from understanding the novelty and the significance of the paper. Indeed, reference [3] is very relevant to the paper, since [3] provides a generic framework for translating online algorithms without delays to those with delays, with a mild deterioration in the regret bounds. The framework of [3] also allows adversarial online learning problems with an oblivious adversary (See their Theorem 1), for example the first problem considered by the authors. While I understand that the authors’ algorithms are different, do the authors’ algorithms suffer a smaller regret bound than those in [3]? There is no discussion on this matter in the main text or in the supplementary material. Such a discussion will allow readers to see the novel aspects of the paper in comparison to [3]. My second concern is about the parameter-free version of Algorithm 1, which is provided in Algorithm 2. In Line 142, the authors claim that the sum of delays can be guessed by the doubling trick shown in equation (8). However, I have a hard time convincing myself on the effectiveness of the learning rate in (8), and I do not think that Algorithm 2 can achieve the regret bound in Theorem 1. Before elaborating on the second concern, I would like to point out that the algorithmic framework in [3] is parameter-free. In order to improve upon [3], the authors need to show that Algorithm 2 has a smaller regret bound than [3]. To understand my second concern, consider the following example. Let say the horizon $T$ is equal to $\sum^L_{\ell = 0} 2^ell $, so that there are epochs $1, \ldots, L$, which are respectively of lengths $2^0, 2^1, …, 2^L$. Suppose that there is no delay in epochs 1, …, $L$, so that $m_t = 1$ in epochs $1, \ldots L$. In addition, suppose that there is say a delay of say $M$ is epoch $L+1$, where $M$ is an integer much larger than 1. On the one hand, at the start of epoch $L+1$ (which is at the start of time $(1 + 2 + … + 2^{L-1}) +1$), the player has observed that $m_t = 1$ for each time $t$ in each of the epochs $1, \ldots, L$. Consequently, the summation $\sum^t_{t=1} m_t$ in the denominator in (8) is chosen as $2^L - 1$. On the other hand, we know that the sum of $d_t$ in epoch $L+1$ (which is required in Theorem 1) is in fact $M \times 2^L $. Given such a discrepancy, Algorithm 2 would not achieve the regret bound in Theorem 1. I could have mis-understood something here, since there is a typo ($\sum^t_{t=1} m_t$) in equation (8), and I take it as $\sum^t_{s=1} m_s$. Regardless, I do not think that the sum of delays can be guessed by the doubling trick. Indeed, different from the horizon $T$ which is increased by 1 per time step with certainty, the sum of delays can grow in an arbitrary manner. There is something missing in moving towards the parameter free version of Algorithm 1. For example, in guessing the sum of delays, there must be some error in the guess, leading to a sub-optimal choice of $\eta$. Can the authors provide a general version of Theorem 1 under an arbitrary $\eta$? Minor Comments. The paper could be reorganized in order to flesh out the novel contributions. The proofs of Corollary 1 and Theorem 2, which occupy much of the main text, only involve routine calculations. The only non-routine part in these proofs are the use of Lemma 3. My understanding is that the novel contributions come from Lemma 3, which is not even stated in the main text, but is stated and proved in the supplementary material. The paper would have been better explained if the authors provide more intuitions on the proof of Lemma 3, and defer the proofs of Corollary 1 and Theorem 2 to the supplementary material. In Line 118, the authors seem to emphasize that that their algorithm 1 is competitive in comparison to the best mixed strategy, which is a tad peculiar since it is clear that the loss under the best deterministic strategy is the same as the loss under the best mixed strategy. In Proposition 1, the authors mention about $l_t$, what is $l_t$? Does $l_t$ equal to $U(\cdot, a_t)$, where $a_t$ is the random action chosen by the column player in time t?

[Author Response · NeurIPS 2019]

**Comparison to existing literature (R1,R3)**: As for **R3**'s major comment, our setting is fundamentally more general than [3], which assumes stochastic and i.i.d. delays, while our delays can be arbitrary. For other literature, [1] assumes a constant delay parameter $d$. [10] considers stochastic rewards and delays. [11] considers the full information case and not the bandit feedback case. It also assumes that all feedback is received before $T$, and that $\sum_{t=1}^{T} d_t$ is known - which we do not assume. **Our paper is the first to address adversarial (arbitrary) delays and costs with bandit feedback**. Additionally, none of them consider zero-sum games with delays, that we show are surprisingly more robust against delays than the single-agent setting. We now include this discussion, with more details.

**Choosing the step size $\eta_t$ when $\sum_{t=1}^{T} d_t$ is unknown (R1,R3)**: We provide Algorithm 2 as an adaptive algorithm that does not require prior knowledge of $\sum_{t=1}^{T} d_t$ and $T$. As shown by the counterexample of **R3**, standard doubling trick epochs are not enough. We now address this issue in detail, fixing Algorithm 2 and providing a full proof that a regret of $O\left(\sqrt{\ln K \left(K^2 T + \sum_{t=1}^{T} d_t\right)}\right)$ is achievable even when $\sum_{t=1}^{T} d_t$ (and $T$) is unknown, using a novel doubling trick. Let $m_t$ be the number of missing feedback samples at time $t$, (including the $t$-th feedback). The idea is to start a new epoch every time $\sum_{\tau=1}^{t} m_\tau$, that tracks $\sum_{\tau=1}^{t} d_\tau$, doubles. Define the $e$-th epoch as $\mathcal{T}_e = \left\{t \,|\, 2^{e-1} \le \sum_{\tau=1}^{t} m_\tau < 2^e\right\}$, with step size $\eta_e = \sqrt{\frac{\ln K}{2^e}}$. Define by $\mathcal{M}_e$ the set of feedback samples for costs in epoch $e$ that are not received within epoch $e$. These feedback samples are discarded once received, and the strategy $\boldsymbol{p}_t$ is initialized at the beginning of every epoch. A compact version of the proof is provided next. The $K^2$ replacing $K$, which has no affect when $d_t \ge K$, can be improved with a more careful computation. To answer **R3**, Lemma 3 is a general version of Theorem 1 for any arbitrary non-increasing $\eta_t$, in particular for any constant $\eta$.

Define $T_e = \max \mathcal{T}_e$, and note that $\mathcal{T}_e = [T_{e-1} + 1, T_e]$. Applying Lemma 3 on epoch $e$ yields

$$R_e \triangleq E^{\boldsymbol{a}} \left\{\sum_{t \in \mathcal{T}_e} \langle \boldsymbol{l}_t, \boldsymbol{p}_t \rangle - \min_i \sum_{t \in \mathcal{T}_e} l_t^{(i)}\right\} \le \frac{\ln K}{\eta_e} + \eta_e \left(\frac{K}{2}|\mathcal{T}_e| + 2\sum_{t \in \mathcal{T}_e, t \notin \mathcal{M}_e} d_t\right) + 2|\mathcal{M}_e|. \tag{1}$$

Now we want to find the maximal $|\mathcal{M}_e|$ such that $\sum_{\tau=T_{e-1}+1}^{T_e} m_\tau \le 2^{e-1}$ is still possible. The "cheapest" way to increase $|\mathcal{M}_e|$ is when the feedback from round $T_e$ is delayed by one (contributes 1 to $\sum_{\tau=T_{e-1}+1}^{T_e} m_\tau$), the feedback from round $T_e - 1$ is delayed by two (contributes 2 to $\sum_{\tau=T_{e-1}+1}^{T_e} m_\tau$) and so on, which gives $\sum_{i=1}^{|\mathcal{M}_e|} i = \frac{|\mathcal{M}_e|(|\mathcal{M}_e|+1)}{2} \le 2^{e-1} \implies |\mathcal{M}_e| \le 2^{\frac{e}{2}}$. Hence, by choosing $\eta_e = \sqrt{\frac{\ln K}{2^e}}$ we obtain

$$R_e \le \sqrt{\ln K}\left(2^{\frac{e}{2}} + 2^{-\frac{e}{2}}\left(\frac{K}{2}|\mathcal{T}_e| + 2\sum_{t \in \mathcal{T}_e, t \notin \mathcal{M}_e} d_t\right)\right) + 2^{\frac{e}{2}+1} \underset{(a)}{\le} 2^{\frac{e}{2}+1}\sqrt{\ln K} + 2^{-\frac{e}{2}-1}|\mathcal{T}_e| K\sqrt{\ln K} + 2^{\frac{e}{2}+1} \tag{2}$$

where (a) follows since every $t \in \mathcal{T}_e$ s.t. $t \notin \mathcal{M}_e$ contributes $d_t$ to $\sum_{\tau=T_{e-1}+1}^{T_e} m_\tau$ (the $t$-th feedback is missing for $d_t$ rounds between $T_{e-1} + 1$ and $T_e$). Therefore $\sum_{t \in \mathcal{T}_e, t \notin \mathcal{M}_e} d_t \le \sum_{\tau=T_{e-1}+1}^{T_e} m_\tau \le 2^{e-1}$. We conclude that

$$E\{R(T)\} = \sum_{e=1}^{E} R_e \le 2\left(\sqrt{\ln K} + 1\right)\sum_{e=1}^{E} 2^{\frac{e}{2}} + \frac{K}{2}\sqrt{\ln K}\sum_{e=1}^{E}|\mathcal{T}_e|\, 2^{-\frac{e}{2}} \le 2\sqrt{2}\left(\sqrt{\ln K} + 1\right)\frac{2^{\frac{E}{2}}-1}{\sqrt{2}-1} +$$

$$K\sqrt{\ln K}\sum_{e=1}^{E}|\mathcal{T}_e|\, 2^{-\frac{e}{2}} \underset{(a)}{\le} 10\left(\sqrt{\ln K} + 1\right)\sqrt{\sum_{t=1}^{T} d_t} + 5K\sqrt{T \ln K} = O\left(\sqrt{\ln K \left(K^2 T + \sum_{t=1}^{T} d_t\right)}\right) \tag{3}$$

where $E$ is the last epoch and in (a) we used that $\sum_{t=1}^{T} d_t \ge \sum_{t=1}^{T} \min\{d_t, T-t+1\} = \sum_{t=1}^{T} m_t \ge \sum_{\tau=1}^{T_E} m_\tau \ge 2^{E-1}$, and also that $\sum_{e=1}^{E}|\mathcal{T}_e|\, 2^{-\frac{e}{2}}$ subject to $\sum_{e=1}^{E}|\mathcal{T}_e| = T$ is maximized when $E = \lceil \log_2 T\rceil$, with maximal length $2^e$ for epoch $e$, so $\sum_{e=1}^{E}|\mathcal{T}_e|\, 2^{-\frac{e}{2}} \le \sum_{e=1}^{\lceil\log_2 T\rceil} 2^{\frac{e}{2}} \le \sqrt{2}\, 2^{\frac{\lceil\log_2 T\rceil}{2}-1}}{\sqrt{2}-1} \le 5\sqrt{T}$.

**Unbounded delays (R1):** We mean that Theorem 2 holds even for **some** unbounded delays s.t. $d_t \le f(t)$ for increasing $f(t)$ (e.g., $f(t) = t\log t$). $f(t) = e^t$, or even $f(t) = t^2$ grow too fast. This is better explained now.

**Ergodic Average (R1):** This is a weighted average that coincides with the standard average for $\eta_t = \frac{1}{T}$. Its importance is mostly just being computable, so a computation of a NE is still possible by sampling/simulating the game even with superlinear delays. When $d_t = 0$, $\eta_t = \frac{1}{T}$ is a valid choice for Theorem 2 which gives the classical result.

**No Exploration Term (R1):** It was shown that the exploration term of the original EXP3 is not necessary (see "Regret Analysis of Stochastic and Nonstochastic Multi-armed Bandit Problems" by Bubeck & Cesa-Bianchi). In any case, our self sufficient proof independently shows that no exploration term is needed. We have now clarified this issue.

**Minor Comments (R1,R3):** We have fixed all minor issues (a-e for **R1**, reorganization and line 118 for **R3**). With some effort, the results are extendable to the continuous case, which is exactly the subject of our current work.

[Meta-Review · NeurIPS 2019]

This paper should be accepted. The initial review was underwhelming. In particular, there was confusion about the tuning of the learning rate and the inner workings of the doubling trick. This was cleared up by the author rebuttal. The reviewers are now convinced of the technical merit, and are favoring accepting the paper. Given the interest in delays and the recent surge in game-theoretic equilibrium arguments/constructions discovered for tackling a variety of problems, I think this paper will interest the NeurIPS participants.